# Implementation of Health Impact Assessment in the Healthcare System of the Republic of Kazakhstan

**DOI:** 10.3390/ijerph20032335

**Published:** 2023-01-28

**Authors:** Zhan S. Kalel, Gabriel Gulis, Altyn M. Aringazina

**Affiliations:** 1Caspian International School of Medicine, Caspian University, 521 Seifullin Street, Almaty 050000, Kazakhstan; 2Unit for Health Promotion Research, University of Southern Denmark, Degnevej 14, Esbjerg 6700, Denmark; 3Olomouc University Social Health Institute OUSHI, Palacky University Olomouc, Katerinska 653/17, 77900 Olomouc, Czech Republic; 4AlmaU School of Health Sciences, Almaty Management University, 227 Rozybakiev Street, 050060 Almaty, Kazakhstan

**Keywords:** Health Impact Assessment, implementation, Kazakhstan

## Abstract

The Health Impact Assessment (hereinafter referred to as HIA) is an effective method for predicting potential health impacts from decisions. Little is known about the implementation of the HIA in the Republic of Kazakhstan (further, RK). In addition, the Russian language literature has not yet been reviewed in terms of HIA-related knowledge. By conducting a literature review of enabling factors, including Russian language literature, on the implementation of the HIA and studying governance systems in RK, we aim to suggest an implementation process to implement the HIA in RK. After careful analysis of the governance system, we suggest set up of a HIA support unit under the National Scientific Center for Health Development and discuss the possible benefits. The proposed center should guide the implementation of the HIA in RK.

## 1. Introduction

Buse, reflecting on the words of Dr. Tedros, Director General of WHO, at the opening of the World Health Assembly 2022 in a recent article calls for a shift in paradigm within existing health systems [1]. One of important suggestions in this article is that “healthy societies are about fixing systems, not people”.

One of key methods to apply in the sense of fixing systems is Health Impact Assessments (further referred to as HIA). [2]

In 1986, the World Health Organization’s Ottawa Charter for Health Promotion recognized the need for health to work across multiple sectors, and in 1997, the WHO Jakarta Declaration called for “equity-based health impact assessments as an integral part of policy making” [3]. The HIA originated at the 1999 Gothenburg Consensus Conference, which formally established the HIA as a methodology to assess future health impacts of recent activities. The methodology continues to improve for wider applications in a variety of issues. for example, tailored for the special needs of the community [4] or specifically for the urban setting [5]. The number of HIAs carried out continues to grow, and interest in it has spread to many countries. HIA can not only improve the final decision but also improve the decision-making process [6,7]. With the participation of those affected by the decision, the HIA makes the decision-making process more open.

The advantage of the HIA is that it creates mutual understanding between different authorities and decision-makers. [8] There are many examples of how local governments and primary healthcare (health authorities) work together on the HIA [9,10,11,12]. Where this has happened, health officials have always had a better understanding of how local governments work and vice versa [2].

Likewise, when organizations and employees unfamiliar with public health become involved in the HIA, they have a better understanding of health issues. Even if the HIA does not influence the decision considered immediately, it is likely that a deeper understanding of the health problems will take hold and influence future decisions, leading to better health [9,13].

Community development is another important public health activity. Community development workers can work with them to learn and understand how their health depends on their environment and living conditions. They then identify how the situation can be changed to improve the health of the population. The HIA builds causal chains to assess how factors will affect health [14].

The HIA is designed to inform decisions and assist decision-makers, so its main benefit should be decisions that are fair and better for health and reduce the risk of making decisions that have unexpected negative health consequences. Determining the health effects of different options should enable decision-makers to find the trade-off needed when choosing between options and allow them to optimize their decision.

Public health in the Republic of Kazakhstan (RK) is on transition from individual, hygiene-based disease prevention to more structural disease prevention and health promotion approach which is fully in line with the suggested paradigm shift. [15] Implementing HIA in RK is therefore an important step within the process. There has been no reports or papers published or made available online about any of the impact assessments carried out in Kazakhstan. The presented paper is exploring the possibility of introducing Health Impact Assessment in the RK and propose a theoretical model of implementation.

## 2. Materials and Methods

To map opportunities for HIA implementation in the RK we used literature review of enabling factors of HIA implementation as well as legislative documentation within the country. The proposed model was analyzed by the first author based on findings from literature. The analysis was supported by consultations with second and third author.

English and Russian language literature was searched on PubMed, Google Scholar, Scopus, and Russian databases—Elibrary and Cyberleninka. Search keywords were restricted to Health Impact Assessment, implementation and the time period covered was 2003–2021.

The legislative documentation analysis was focused on demonstrating the main outline of Kazakhstan’s state system and how decision-making process in public health area is organized in it. The search for official documents was carried out on two biggest and timely updated databases for legal acts: Information and legal system of regulatory legal acts “Adilet” and information system “Zakon”.

Inclusion criteria:Considers the model of implementation of HIA into existing systems;Demonstrates how HIA is currently established and functions in the country;Has full text available for free;Not older than 2003.

## 3. Results

### 3.1. Implementation of HIA on International Level

Figure 1 presents the flow chart summarizing the number of articles identified through the literature review.

According to the literature search, the main factors that contribute considerably to create a strong basis for the HIA were identified: existence of a policy in the HIA, integrated infrastructure, and capacity building. Sources that either identified such factors as enabling directly or highlighted the benefits of their influence indirectly were taken into account.

The policy on the HIA refers to a supportive legal basis to form a framework for conducting HIAs at the local or national level. The policies might include determining a specific responsible body, the level of its juridical power, integration with health in all policy approaches, clear vision of the policy, and funding sources [14,16,17,18,19,20,21,22,23,24,25,26,27,28,29,30].

Integrated infrastructure includes factors related to an interconnected network of legal bodies, non-governmental organizations, and community, such as relationships with sectors, with key decision-makers and locals, involvement of field experts, and data availability. These factors enable the policy cross-sector collaborations to provide multiple forms of evidence and perspectives [31,32,33,34].

Capacity encompasses the factors that increase the competency and overall readiness to enact those policy requirements and use the resources at hand to their full potential. It is experts trained in the HIA, development of programs, briefings and materials, the amount of available finances, and its management [35,36]. The development level of the current institutional infrastructure and capacity to grow in this direction serve as a fertile ground for the stable implementation and strengthening of HIA policy.

The research conducted in the European region during Phase IV of the World Health Organization European Healthy Cities Network, in which HIA was one of the core themes, showed a number of other factors able to facilitate the development of HIA: resources provided by the WHO and the HIA subnetwork in the form of documents, membership of a network, and willingness or interest on behalf of politicians and technicians to be involved with HIA implementation [34].

There are different approaches to implementing the HIA. It can be loosely classified into several parameters that can define an HIA system.

First, it differs based on two main aspects: Is it compulsory or voluntary. Are projects, policies, and programs legally obliged to undergo HIA? Quebec Province, Canada has a mixed system, where the Department of Health and Social services is responsible for assessing the health impacts of legislative provisions and regulations proposed by the government while, at the municipal level, the HIA remains voluntary [29].

The second big key characteristic is the levels HIAs are conducted on. Is it national or local?

For example, if it is regulated officially by the government, it could be monitored by the highest political structures, including ministries or their departments. In some countries, it is regulated by the constitution [37].

In France, assessments are funded by the Regional Health Agencies and done by the experts in the field from universities and health observatories [29]. Denmark is another example of HIA carried out on a municipal basis [38].

Wales is one of the prominent examples of the implementation model for the HIA: the Health Impact Assessment Support Unit (WHIASU) was established as part of a broader strategy to improve health and reduce inequalities funded by the Welsh government [39].

Third, if they are carried out in the form of a pure HIA, solely focused on health impacts, adapted from or as a part of another complex assessment.

In Myanmar and Cambodia, the health considerations are planned to be integrated into the existing EIA [37], as it is established in such countries as Mongolia [40], Czech [41], Lesotho [42], Republic of South Africa [43], or Australia [44]. This method is legally supported recently also within countries of the European Union initiated by a Directive of European Commission (Directive 2014/52/EU). The Republic of Cameroon is gradually starting to incorporate the HIA, while, historically, it already developed a strong policy basis for the EIA, regulated by the Ministry of Environment and the Protection of Nature [45]. Visual representation is shown in the Table 1.

To the contrary, Russia’s attempt to implement the EIA has failed, theorized to have happened due to established preexisting forms of state ecological expertise [46], while other types of assessments are not carried out at all. The majority of the studies found during the literature search were one-dimensional environmental risk assessments. The closest studies were mainly prognostic impact assessments [47]. In Thailand, the HIA is adapted as a community HIA (CHIA) [48].

A number of countries are still in the process of evaluating the importance of HIA and its implementation model [49]. The biggest percentage of the participants in the research conducted in Spain voted for mandatory HIA, mostly related to the belief that HIA implemented by statute has bigger power and will be applied more frequently [50].

**Table 1 ijerph-20-02335-t001:** Types of implemented HIA systems.

	Adapted HIA	Standalone HIA	Part of / In Conjunction with Another Assessment	Regulation
Countries with HIA	Thailand—CHIA [48]	Wales [33]; Spain [50]	Mongolia—EIA [40]; Czech—EIA [41]; Lesotho—EIA [42]; Republic of South Africa—EIA [43]; Australia—EFHIA [44]	Nationally
		France [29]; Denmark [38]		Locally
		Quebec [10]; Sweden [24]		Mixed
Being implemented		Iran [49]	Myanmar, Cambodia, and Cameroon—EIA [37]	

### 3.2. HIA and Decision-Making in RK

To consider the implementation of HIA in RK, it is necessary to understand the sociopolitical context and existing decision-making system of the RK. All measures and actions taken at the republican, regional, and local levels of government related to health are under the control of the centralized state system. Research of the RK legislation includes 11 laws found to be relevant to our goals.

Decisions at the state level that have legal force are laws, regulatory legal acts (hereinafter RLA). These acts must be carefully thought out and agreed upon.

The RK is a unitary state with a presidential form of government. The main political directions in the development of the state, including in the field of healthcare, are determined by the president through the Annual Messages to the people of RK, the approval of the strategy and strategic plan for the development of RK, and state programs for the development of the sectoral and intersectoral levels. The development of the main directions of state policy in the field of healthcare is carried out by the government, which approves the state program for the development of healthcare. The Ministry of Health (hereinafter the MH) implements the state health policy [51].

Legislative functions are performed by the Parliament, which consists of two chambers: the Senate and the Mazhilis, acting on a permanent basis. At a joint meeting of the chambers, the Parliament has the right, upon the proposal of the President of the RK, to introduce amendments and additions to the constitution. The Parliament, in a separate session of the chambers, adopts constitutional laws by the successive consideration of issues first in the Mazhilis and then in the Senate [52].

The government exercises executive power, heads the system of executive bodies, and manages their activities. The government, in turn, is subordinate to the Prime Minister, and one of its executive bodies is the MH [53].

Figure 2 shows the state system of the RK.

Directly in the process of developing policies and management decisions in the field of healthcare at the sectoral level, both specialists of the MH and organizations at the republican level are responsible for the development of the relevant specialized service, various collegial bodies created under the ministry with the involvement of key stakeholders, representatives of civil society on an ongoing basis and temporarily (working groups). On an ongoing basis, the practice is placing a state order by several ministries for conducting analytical and sociological research in areas relevant to policy development. When developing projects, the Ministry sends the RLA for approval within the framework of their inherent competencies to subordinate organizations and publishes them on the electronic web portal for 15 days, where every citizen can consider the proposal and provide their recommendations [54].

The MH approves the orders of the minister and can also submit for signature a draft resolution of the Government of the RK to the Prime Minister without the approval of the Mazhilis and the Senate of the Parliament. Bills, which are signed by the president, pass through the consideration of the Mazhilis of the Parliament.

Depending on the goals set for achieving the health development indicators, the ministry has the right to develop its own proposals and programs required to fulfill the tasks set by the state strategy.

Under the MH, a public council is formed, which is an advisory body. The public council discusses the draft budget programs of the ministry and the draft strategic plan, as well as the state and government programs. The main direction of the Council’s activities is work on draft regulations affecting the freedoms and responsibilities of citizens in the field of healthcare [55].

The second way of implementing decisions is carried out through scientific and technical programs (hereinafter STP). Independent experts have the right to implement their proposals and solutions based on this system. Proposals are submitted to a competition in which they can win a grant, which implies financial support for the work on the implementation of their proposal. Factors such as the relevance and effectiveness of the proposal play a significant role.

A scientific and technical project or program is a document that includes the content of the proposed scientific and technical work, representing a scientific and/or technical experimental design and marketing research with a justification of the goal and objectives, relevance, novelty, scientific and practical significance, and feasibility carrying out the planned work. The main principle of the implementation of scientific and technical progress is to stimulate and support the priority areas in accordance with national interests, the needs of the socioeconomic development [56].

When working with STP, the MH cooperates with the Ministry of Education and Science. The MH announces a competition for scientific and technological propositions within the framework of the program-targeted financing government decree. After submitting documents for the competition, the proposal undergoes an expert assessment [57].

Independent state scientific and technical expertise of projects and programs is the most important mechanism of the program-targeted method [58]. During the examination, the scientific validity of the program, the optimal size and composition of the research team, the reality of achieving goals based on research projected by the parent organization, the need to attract co-executing organizations, etc. are checked and evaluated. When choosing a parent organization for fundamental research programs, the criteria are the presence of a strong scientific school, recognized scientist leader, and possession of a modern research methodology [59].

Then, this proposal is considered by the National Scientific Council, consisting of Kazakhstani and foreign scientists. Councils make decisions on grants and program-targeted funding (or the termination of funding), at the expense of the republican budget [60].

### 3.3. Model of Implementation in RK

Where and how the RK HIA unit can be created is the next important aspect that needs to be studied. A representative of this body can serve as an HIA expert in the councils for the consideration of proposals for RLA or STP. In theory, it will consist of experts from various research centers and universities of the corresponding area and will be under the wing of one of the organizations at the Republican level for the development of healthcare.

If the implementation of compulsory HIA on the basis of legislation is considered, then it will be necessary to develop regulatory legal acts at the state level for the centralized, controlled monitoring of the implementation of a HIA for each project or program affecting health.

One more factor to keep in mind is that HIA is most optimally used at the stage prior to making a decision. This allows the timely identification of possible adverse impacts on the health and assessment of environmental consequences. Discussion of the results of the assessment with the public and the expert community will allow taking into account public opinion and developing measures to reduce and prevent impacts.

What we propose is the creation of an HIA unit under the National Scientific Center for Health Development, where it will be in the optimal position to acquire needed recourses and have access to the local experts and data, as well as connections with other fields supported by the MH. The center has made a great contribution to the development of the domestic healthcare system, participating in the development and methodological support of the priorities of state programs, roadmaps, industry projects, and various legal acts. It is also the Center for the Activities of the Joint Commission on the Quality of Medical Services, the Body of the Formulary Commission, expert and analytical support for their work, and the National Coordinator for Human Resources in the field of healthcare. The center also cooperates with the World Health Organization, the US Centers for Disease Control and Prevention, the International Society for Quality in Healthcare, and the International Society for Health in Healthcare. The center was established in 1994 in Almaty.

The schematic representation is demonstrated in Figure 3 below.

Since the decisions implemented in the RK go through many stages, departments, and inspections, it is difficult to determine when it is suitable for the HIA. It makes sense to integrate the screening process into one of the councils (Public Council for RLAs and independent state scientific and technical expertise for STP), while the dedicated unit carries out latter stages of the assessment.

To sum it up, we believe our model proposes an optimal position that meets the criteria identified on the basis of a literature review: strong policy support from MH, good entry point into the decision-making process, availability of data and experts that can undergo training or guided pilot assessments through local departments, and collaboration with other ministries, a strong position within the infrastructure.

## 4. Discussion

One of the requirements for a fair assessment is the voluntariness of passing the assessment, but, on the other hand, for its implementation, a certain legislative basis is required. Making the assessment mandatory for all will potentially open up new channels of corruption or make the assessment process a mere formality, not to mention the need for a body to screen each decision for the need for a HIA. Without a regulatory framework, the methodology will not function at a sufficient level and risks remaining unused at all. Therefore, a well-thought-out implementation must strike a balance between the two.

Undoubtedly, one of the most important factors in the implementation of HIA has been policy support. There were no examples of HIA being effective and consistent without at least some kind of legislative framework. We felt that was the strongest option for creating a legal framework to create an HIA unit, even if as a temporary measure. It will be able to better understand its needs, adapt, and formalize the relevant standards for the RK while receiving methodological guidance from other more experienced states. The governmental structure of the RK is very top-down and centralized. Therefore, it is logical to make HIA regulated officially for it be taken seriously, which led us to consider other approaches than those in Denmark or France.

In parallel with Thailand’s Constitutional basis for HIA, the legal and regulatory framework for the implementation can be included in the national code “On people’s health and the health care system”, which is updated and supplemented as the main document governing such issues as government regulation, management, control, and supervision in the field of healthcare and public health protection. Over the past decade, the RK has been working to streamline the existing laws on health and medical care to create a single, unified body of laws. When developing state policy in the field of health, RK is guided by international agreements and programs in which it participates, in view of their compliance with the goals set by RK in the national strategy “Kazakhstan-2050”. Taking into account the impact that HIA can have as a tool to support decision-making in favor of health, work towards sustainable development goals, the Healthy Cities program, of which the RK is a member of, and then the introduction of the HIA as a law, can be justified.

In 2021, the RK introduced an EIA with the Environmental Code [61]. This is the first form of an impact assessment in the RK, although, previously, it was based on standard hygiene-based environmental risk assessments, as in Russia. The EIA is still relatively new, and the inclusion of health may overly complicate the methodology at this early stage. Although it includes “the health and living conditions of the population” as the subject of the assessment, it is tertiary in relation to environmental and wildlife impacts. For a successful integration, the two RK would need EIA to have strong foundation in the first place, as demonstrated by Quebec or Cameroon. This approach of gradually combining one impact assessment into another has its advantages, namely cost-efficiency and having a more organic, grounded development into complex forms. The reason why we turned down this way of implementation is twofold: (1) a proper standalone implementation will increase the risk of reports being incoherent and low quality, while a combination might lead to confusion and decrease in the understanding of what the assessment is trying to do, and (2) a history of a hygiene-based approach to public health in theory has already laid some groundwork for Kazakhstan experts to grasp the principles of the EIA, therefore eliminating the need to ease the introduction of a new impact assessment through combinations.

A good example of a functioning HIA unit—the aforementioned WHIASU, which provides guidance, training, resources, and information in relation to the practice of HIA. The executive power is a characteristic of both Welsh and RK governments that can potentially serve as the key to a functioning unit with access to key stakeholders, resources, and a healthy level of authority. A good balance can be achieved by adapting one of the mixed approaches: on the level of policies and national programs, screening would naturally fit into council examinations, while on the level of local projects and proposals, the HIA can remain voluntary for decision-makers willing to improve upon them and build a good reputation in the public eye.

### Limitations

The main limitations of this study were recourse-related.

Only free publications being accessed during the literature review potentially dismisses a considerable part of acquired scientific knowledge. However, the effort has been made to counteract this limitation by extending the researched time period within the framework that was considered to be possible to cover.

A number of included researched keywords was limited due to our intention to narrow down the search criteria to discover publications that are focused on ways to implement HIA or, at the very least, reports that mention the enabling factors of the conducted assessment. We felt that other synonyms for “implementation” did not represent exactly what we were looking for.

There were no economic efficiency calculations of the proposed model made. In the context of the Kazakhstani system, most propositions are accompanied with such calculations and budget-planning if they strive to be taken seriously at the state level. Unfortunately, the estimation of required financial recourses and potential return was too big of a task for the scope of our research.

The categories of enablers for HIA implementation were simplified and condensed into three main factors while arguably blurring the nuance and specificity. The streamlined approach for the sake of the clarity and relative simplicity of the theorized model might lead to some factors getting overlooked in favor of others. None the less, we believe that the process of implementation, whatever form it takes, will benefit from the clearer vision, taking into account a plethora of unforeseen consequences and obstacles that will arise with high probability, and will be better dealt with on a case-to-case basis.

## 5. Conclusions

RK is in a state of gradual transformation fueled by the strive for development. Big paradigm shifts in how Kazakh systems perceive and operate are opportunities not to be lost. Our model of implementation based on the creation of an HIA unit under the wing of the MH can serve as a starting point for the integration of health concerns into decisions at all levels, as well as a valuable addition to vitalize the discussion around adapting and applying new methodological tools in RK and Central Asia.

## Figures and Tables

**Figure 1 ijerph-20-02335-f001:**
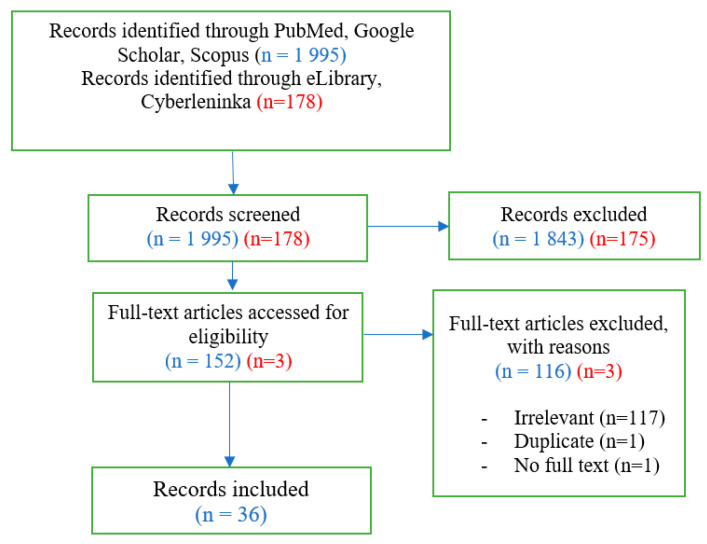
PRISMA flow diagram. The red code indicates the English scientific databases; the blue code indicates the Russian databases.

**Figure 2 ijerph-20-02335-f002:**
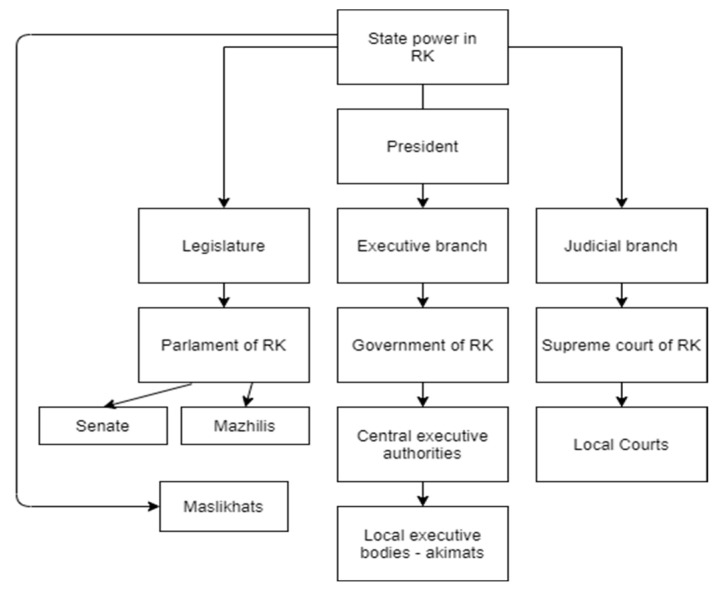
State power hierarchy of RK.

**Figure 3 ijerph-20-02335-f003:**
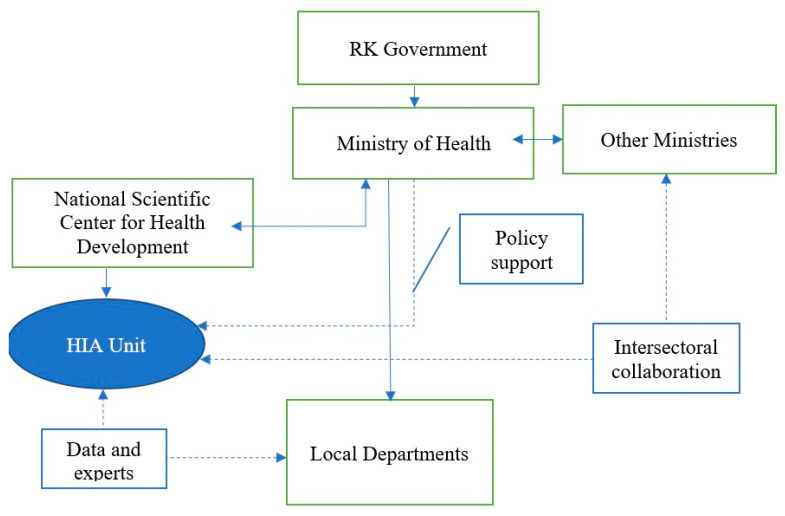
Model of HIA implementation.

## Data Availability

The Kazakhstani regulatory legal acts used to support the findings of this study are available from the corresponding author upon request.

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
