# Peer review of "Implementation of Health Impact Assessment in the Healthcare System of the Republic of Kazakhstan"

_ijerph, 2023, doi:10.3390/ijerph20032335_

Round 1

Reviewer 1 Report

Thanks for this paper which I read with interest. It has a nice style and structure. However, I do believe it needs some revisions before being published. 

The Introduction needs more referencing as there are several statements which are not sufficiently backed up by references and are generalisations. The Introduction should also highlight that there are challenges and benefits to the introduction and implementation of HIA. Finally, there is no background on the use of impact assessment including HIA in RK to date.

Materials and methods

Can you say why the key words were so few and those years were chosen? Also, I understand that this review was at an international level but did you not look for any HIAs conducted in RK at all? 

line 156 - did the spanish participants say why?

line 234 - 236 - so how and where could HIA fit into these decision making processes? 

line 240 - Why academia? why research and experts? most HIAs are not conducted by academics but out in the field by public health practitioners and local government officers across settings and sectors or private consultants. 

line 253 - is this an academic centre in the MH?

line 272 - this is a dangerous thing to do!! There are issues and challenges around this - you need to be clear about how to manage expectations that this Unit will do HIAs - or this that the intention? 

line 290 - like? what about Wales?

line 303 and 304 - this should have come sooner - any other IAs carried out in RK? 

line 319 - you may wnat to say a bit more about the welsh model

Reviewer 2 Report

The proposed theme is interesting and the review of the literature is obviously useful to have an overview of the different international experiences but the work has strong methodological limitations that make the result uncertain. When organizational theoretical models of an activity are presented, it is necessary to proactively evaluate the criticalities with all the interested parties before being implemented. In particular, it is necessary to answer the question, how much will this model be able to work once applied? In particular, the issue of the HIA presents numerous criticalities due to the many actors (politicians and stakeholders) who can influence the final result of the evaluation; the main difficulty of the institutional or scientific bodies that carry it out is to maintain a coherent line of conduct. Furthermore, in complex systems such as healthcare, it must also be said that solutions that have proven themselves valid in one context need not be valid in another.

It is necessary for the publication of this paper better to define its purpose. What is its main aim? A literature review on HIA modalities in various countries or a proactive analysis of a single model (model of implementation in RK). The authors seem to use the results of a literature review to define a model but It's too simplistic modality to achieve the goal. I would suggest to carry out a complete and good review and as a second step a proactive analysis of the model.

Better defining the purpose is useful in relations to methods of analysis. The Prisma method is used for selecting the experiences and understanding what is done internationally. On the basis of this information, an RK model is proposed which is then analyzed to evaluate its implementation. It would be necessary that the methodology describes not only the methods of the review but also how the analysis of the proposed model took place (focus group or others).

The first part of this paper (review) on how HIA has been implemented internationally should be summarized in a couple of tables to make it easier to compare the selected papers, in the current way we are limited to comments but not the compliance with criteria.

With regard to the model proposed for RK, it is necessary to bring out the organizational criteria adopted on the basis of the review and therefore to better clarify the factors facilitating or hindering its implementation according to a proactive analysis.

Some clarifications. Why did you identify the inclusion and exclusion criteria, wasn't it enough to declare the inclusion ones considering that the exclusion ones are the exact opposite?  A shorter period of time could not have been considered, why was 2003 chosen? Moreover to include for the review only the free papers is an unfortunate limitations.  The abstract doesn't fully explain the work done.

Round 2

Reviewer 1 Report

This is much better and I like the table.

I would like this to be added in regardless of space limitations! It is important:

line 156 - did the Spanish participants say why?

This is mostly related to believe that HIA (or any other method) implemented by statute has bigger power and will be applied more frequently. Due to space limitation, we did not comment more on this issue in our manuscript.

Also, I do not see where the additional refs have been added into the track changes nor reference list. Please add these or highlight them. 

Author Response

This is mostly related to believe that HIA (or any other method) implemented by statute has bigger power and will be applied more frequently. Due to space limitation, we did not comment more on this issue in our manuscript.

I would like this to be added in regardless of space limitations! It is important:

line 156 - did the Spanish participants say why?

The information was added and highlighted in an updated manuscript.

Also, I do not see where the additional refs have been added into the track changes nor reference list. Please add these or highlight them. 

I am sorry, the issue was with my reference tracking software, problem fixed and updated in a manuscript. 

Thank you for your thoughtful comments!

Reviewer 2 Report

The answers of the authors do not completely satisfy me even if small improvements have been made and the methodological limitations have been clarified. However while continuing to present the study of the shortcomings it may be of interest to the readers.

Author Response

The answers of the authors do not completely satisfy me even if small improvements have been made and the methodological limitations have been clarified. However while continuing to present the study of the shortcomings it may be of interest to the readers.

Limitations subchapter has been expanded upon in the updated manuscript.